# A Short Form of the Child/Youth Health Care Questionnaire on Satisfaction, Utilization, and Needs in Children and Adolescents with a Chronic Condition (CHC-SUN-SF/YHC-SUN-SF)

**DOI:** 10.3390/ijerph182212196

**Published:** 2021-11-20

**Authors:** Holger Muehlan, Henriette Markwart, Ingo Menrath, Gundula Ernst, Ute Thyen, Silke Schmidt

**Affiliations:** 1Department Health & Prevention, University of Greifswald, 17489 Greifswald, Germany; henriette.markwart@posteo.net (H.M.); silke.schmidt@uni-greifswald.de (S.S.); 2Department of Child and Adolescent Medicine, University of Luebeck, 23538 Luebeck, Germany; ingo.menrath@uksh.de (I.M.); ute.thyen@uksh.de (U.T.); 3Medical Psychology Unit, Hannover Medical School, 30625 Hannover, Germany; ernst.gundula@mh-hannover.de

**Keywords:** health care satisfaction, chronic condition, children, adolescents, child health

## Abstract

We decided to develop a short-form of the CHC-SUN/YHC-SUN, a questionnaire aiming at assessing health care satisfaction of children and adolescents with chronic health conditions. Data analysis was based on samples from three different studies. Item selection involved statistical analysis and expert consensus. For independent validation purposes, we calculated descriptive statistics on single-item and composite-scale levels and applied classic test theory, confirmatory factor analyses, and correlation analysis to investigate the psychometric properties of the final short-form by different types of reliability and validity. Internal consistency (Cronbach’s Alpha) reached values of a = 0.89 (self-report) and a = 0.92 (parents report), split-half reliability values reached 0.85 (self-report) and 0.91 (parents report). Confirmatory factor analysis indicated no sufficient fit for the single factor solution, whereas the solution with three factors and one higher order factor indicated the best overall fit amongst three competing models. Validity of the short-form measure can be assumed, e.g., as indicated by its association with a single-item measure on general health care satisfaction. The short-form measures of the CHC-SUN for parents (CHC-SUN-SF) and the YHC-SUN self-report version for adolescents (YHC-SUN-SF) feature excellent psychometric performances, provide economical assessments, and are easy-to-administer questionnaires. They should be used whenever brief measures are needed for economic reasons.

## 1. Introduction

Throughout their development, children and adolescents with chronic conditions face continuous challenges with respect to the impact of their conditions on various domains of their daily life. Therefore, developmental science and pediatric care has been paying increased attention to the well-being of this particular vulnerable group of patients. Resulting from this, research suggests that the well-being of these patients could be positively influenced in the early phases of medical treatment, which in turn leads to improved adaptation, increased participation, and thus a more fulfilling life and a higher quality of life [1]. Therefore, identifying special needs of children and adolescents with chronic health conditions is of great importance, as the health status and well-being of young patients has an impact on both childhood and adolescent development [2] as well as the quality of life of the entire family [3,4]. Developing reliable and valid instruments for the assessment of patient-reported outcomes or experiences such as quality of life could make a significant contribution to any activity directed to improving their well-being [5,6]. Incorporating such patient-reported outcome or experience measures in pediatric care would finally support those under treatment and their families—but also their respective health care professionals [6]. It would enable them to gather more information about the treatment of chronic conditions, the effects of different therapies on care processes, as well as their outcomes as seen from the patient’s point of view [7].

Beside health-related quality of life and disease-specific quality of life, another patient-reported outcome that is receiving increasing attention is satisfaction with health care. However, in pediatric care, in contrast to adult medicine, there has been a lack of research on this outcome and its use in clinical practice [7]. This could be because assessing patient-reported outcomes in children and adolescents is much more complex than in adults, since the specific characteristics of this population (e.g., cognitive development) as well as of the pediatric setting (e.g., non-specialized care) make it necessary to account for these factors in constructing appropriate measures [6]. Therefore, previous studies frequently used proxy-reported assessments from adults such as parents or teachers. However, despite these challenges and obstacles when applying self-report measures in the assessment of children and adolescents, it is strongly demanded to take care of the patient’s perspective as provided by patient-reported outcome measures. In addition, there is a need for short and easy-to-administer instruments measuring satisfaction with health care, especially for the assessment of children and adolescents [6]. Finally, when assessing satisfaction with health care, the multi-dimensionality of the construct has received little attention in pediatric research yet and should be taken into consideration whenever possible as well. In sum, these issues lead to shortcomings in the implementation of patient-reported outcome measures in pediatric clinical care and pediatric health services studies, setting the scene for instrument development for this domain within this context.

To account for those challenges, the “Child Health Care–Satisfaction, Utilization, Needs” (CHC-SUN) [8] questionnaire was developed, assessing special health care needs and satisfaction with health care services in children and adolescents with chronic conditions. Initially, the measure exclusively was tested and made available as a parents report from the perspective of parents/relatives. In addition to the initial parents report version of the instrument, it was adapted and validated as a self-report version for adolescents as well and was named “Youth Health Care–Satisfaction, Utilization, Needs” (YHC-SUN) [9]. Due to the length of the original measures [8,9], we decided to develop a short-form, aiming at reproducing the core components of the second “satisfaction” module as closely as possible. Thus, the aim of the current paper is to document the development of these short-form versions and provide evidence for the psychometric quality of the short-form measures of the CHC-SUN for parents and the YHC-SUN self-report for adolescents (CHC-/YHC-SUN-SF) as indicated by different types of reliability and validity.

## 2. Materials and Methods

### 2.1. Measures Applied

Health care satisfaction, utilization, and needs: Both versions (self-report, parents report) of the original long-form measure of CHC-/YHC-SUN were applied. This measure consists of 40 items in total, assigned to two different core modules. The first module includes 14 items and refers to the use of and the demand for health care services (“provision of services”), with special emphasis on the assessment of unmet special health care needs. The second module relates to “satisfaction with care” and covers the subjective perception and evaluation of the quality of the provision of health care services. This module consists of 6 different scales with 25 items and an additional single-item measure assessing satisfaction with health care in general. As opposed to this single-item measure, the six scales relate to specific domains of health care, namely, “diagnosis/information”, “doctor’s behavior”, “child-centered care”, “coordination”, “hospital environment”, and “school” (cf. Figure 1). Internal consistencies of the subscales range between α = 0.80 and 0.95 for the CHC-SUN and between α = 0.75 and 0.93 for the YHC-SUN. Items were selected based on content-related expert rating as well as psychometric criteria. Validity was demonstrated with respect to construct, content, and discriminant validity [8,9].

Health-related quality of life: Health-related quality of life was assessed using an index of the Short Form Health Survey with 8 Items, the SF-8 index [10,11]. The items included in this measure are not directly derived from the Short Form Health Survey with 36 items, the SF-36 Health Survey [12,13]. Each of the 8 items is considered to represent the main content of one of the eight subscales from the SF-36; thus, every item serves as an indicator for one factor of the original scales. Scoring procedures recommend the computation of scores for two summary scales (physical/mental), both also included in the SF-36 and SF-12 Health Survey Measures. Internal consistencies of the summary scales reach α = 0.78 and α = 0.87.

General quality of Life: An index derived from the World Health Organization Quality Of Life (WHOQOL) measures was developed and validated within the European Health Information System (EUROHIS) project and is referred to as “EUROHIS-QOL” [14,15] or “WHOQOL-8” [16]. It serves as a generic measurement, assessing subjective global quality of life in the general population. Although it aims to assess one dimension with respect to psychometric properties, it conceptually represents items derived from different domains of quality of life represented within the WHOQOL measures. The WHOQOL-8 reveals sufficient psychometric properties, e.g., with respect to internal consistency of the scale (α = 0.86; Power, 2003).

### 2.2. Outline of the General Analytic Strategy

Based on expert consensus, it was decided to construct both versions (self-report, parents report) of the short-form as close as possible to the original CHC-SUN and YHC-SUN [8,9] in terms of psychometric criteria especially related to short-form development, such as conceptual representation, statistical association, psychometric comparability, and economy/efficiency [17]. Furthermore, the resulting item pools should display sufficient psychometric properties as stand-alone fixed-length short-form measures. Item selection was performed not only using psychometric indicators, but also by combining a rationale and a statistical approach to instrument construction. To validate the extracted item pool for use as a short-form measure, we conducted the following analytical steps:

Analyzing descriptive statistics: Items and composite scale were evaluated regarding missing data rate and basic descriptive indicators, such as mean and standard deviation, skewness, floor and ceiling effects. Additionally, psychometric performance on item level was investigated by calculating item internal reliability (corrected item–total correlation).

Testing reliability: Internal consistency and homogeneity of the measures were identified by Cronbach’s alpha coefficient. Split-half reliability was computed as well. Concordance between self-report and parents report assessments was investigated using intraclass correlation coefficients (ICC).

Exploring validity: Factorial validity was investigated by applying confirmatory factor analysis (CFA) with maximum likelihood parameter estimation, comparing three a priori defined competitive models, delineated from the measurement model of the original version. We selected the model with the best fit statistics, assuming the standardized root-mean-squared residual (SRMR) close to 0.08 or below, the root-mean-square error (RMSEA) close to 0.06 or below, and the comparative fit index (CFI) close to 0.90 or higher as meaningful cut-off criteria according to the literature [18,19,20].

Hypotheses tested: Other validity estimations were performed by calculating Pearson correlation coefficients for investigating the hypothesized association between composite CHC-/YHC-SUN-SF scores and indicators of generic satisfaction with health care (convergent validity), assumed to be highly associated with the short-form score, as well as general and health-related quality of life (divergent validity), both assumed to be just low or moderately associated with satisfaction with care.

### 2.3. Statistical Software

Confirmatory factor analysis was processed by M-PLUS software version 7 [21]. Further standard statistics were computed using the IBM SPSS version 22.

### 2.4. Data Samples

According to the steps in short-form development recommended in the literature [22,23], data analyses for item selection and instrument validation were performed based on independent samples. (i) For item selection purposes, samples from the multicenter DISABKIDS [24] and TRANSITION [25] projects served as data bases (“extraction samples”), the DISABKIDS study sample for the parents report version and the TRANSITION study sample for the self-report version. (ii) For instrument validation purposes, the sample of the multicenter MODUS project [26] was used (“validation sample”), as in this MODUS study sample, for the first time, both short-form measures (CHC-/YHC-SUN-SF) were simultaneously applied to all study participants and their parents.

Samples were recruited by various collaborating centers within the different projects. Whereas the TRANSITION project and the MODUS project were conducted as multicenter studies by national consortia within Germany, the international DISABKIDS project included seven European countries (Austria, France, Germany, Greece, The Netherlands, Sweden, and United Kingdom). Thus, for the purpose of the present study, regarding the DISABKIDS study sample, we just selected a subsample with young patients from Germany and Austria for data analyses to match this sample with the other two samples regarding language. Consequently, all included patients and relatives were German speakers, and therefore all measures applied were provided in German language. Moreover, all samples included in item selection and validation analysis were restricted to adolescents and young adults 14 years old or older.

## 3. Results

### 3.1. Item Selection

For the development of CHC-/YHC-SUN-SF measures, we decided, by expert consensus, to restrict item selection to three of the six original scales (“diagnosis/information”, “child-centered care”, “doctor’s behavior”). These scales proved most relevant to the construct as indicated by psychometric analyses, conceptual level, and expert consensus (cf. Figure 1). The scales correlate substantially with one another, all of them assessing satisfaction with care at the very heart of health service provision. This is obvious, given that the content of the items directly relates to the interaction between doctor and patient. As opposed to this, the other three scales (“coordination”, “environment”, “school”) do not immediately reflect the provision of health care services but represent more “contextual” domains of health care delivery, e.g., the coordination of services or the facilities of the hospital environment.

In addition to the restriction of scales, we decided to include at least two items of each scale, with an equal number of items from each scale. In principle, this would enable us to estimate approximate values for the three core domains of satisfaction with health care. Moreover, as compared to other options, this alternative was assumed to provide the best efficiency defined by the optimal ratio of economy gained to information retained, to rationally justify this short-form development [23]. Thus, 18 items were deleted, i.e., 75% of the original item pool (cf. Figure 1). After considering these preconditions for this short-form development, we empirically conducted item selection by means of psychometric analysis. Methodologically, these analyses cover the inspection of the item correlation matrix and the computation of reliability estimators. Regarding the psychometric properties, the item pool should have reached a high consistency (>0.80) to ensure a minimum of required homogeneity [27]. On item level, corrected item-total correlations should have reached a value > 0.40. Different solutions for item selection matching these psychometric properties were provided. All items included were rated by three members from the developer’s team of the original measure, with regards to their relevance, independently. Finally, we decided for the item solution with most items that were frequently judged as “indispensable”. Data analyses were based on both “extraction samples”, including the data set of the “DISABKIDS” project field study (*n* = 1.152) for the parents report version and the data set of the “TRANSITION” project pilot study (*n* = 372) for the self-report version of the original CHC-/YHC-SUN. Thus, we developed both short-forms for these versions of the CHC-/YHC-SUN-SF simultaneously. After inspection of the initial analysis based on the data from both “extraction samples”, it turned out that even a minimal selection of two items per scale, and thus of six items in total, would provide very good psychometric properties for the short-forms under construction.

### 3.2. Validation Analysis

#### 3.2.1. Sample Characteristics

For validation purposes, our “validation sample” (*n* = 484) served as a database for the final psychometric analysis. Overall, 355 participants filled out the CHC-/YHC-SUN-SF on their own (self-report) as well as through their parents (parents report). Moreover, 114 participants just answered the SF, and for 15 cases, only the parents report was available. The mean age was 18 years (SD = 1.54, range = 14–24), 52% of the participants were female. At the time of the survey, the majority of the participants (71.7%) went to school, 14.7% were in occupational education, 4.1% studied, 2.7% were employed, 0.8% were unemployed, and 6% provided no information or their current activity was coded under miscellaneous. The sample included patients with a diversity of chronic conditions, with diabetes (19%), hyperactivity (14%), inflammatory bowel disease (14%), asthma (11%), and atopic dermatitis (8%) being the most frequent ones.

#### 3.2.2. Descriptive Analysis

We inspected the data set applying box plots for outliers, but no extreme values were apparent. The Kolmogorov–Smirnov test indicated a significant deviation from a normal distribution for both self-report and parents report (all *p* < 0.001). Descriptive item characteristics of the self- and parents report versions of the CHC-/YHC-SUN are provided in Table 1a,b.

#### 3.2.3. Factor Structure

For confirmatory factor analysis, we applied maximum-likelihood parameter estimation. The results (cf. Table 2a,b) indicated a lack of fit for the single factor solution (self-report: CFI = 0.91; RMSEA = 0.18; SRMR = 0.07/parents report: CFI = 0.84; RMSEA = 0.30; SRMR = 0.09), whereas the solution with three different factors with a higher-order factor indicated the best fit for both self-report (CFI = 0.99; RMSEA = 0.06: SRMR = 0.02) and parents report (CFI = 0.99; RMSEA = 0.09: SRMR = 0.02).

#### 3.2.4. Reliability

Internal consistency values (Cronbach’s Alpha) reached 0.89 (self-report) and 0.92 (parents report), and split-half reliability values reached 0.85 (self-report) and 0.91 (parents report), with sufficient internal consistencies of both test-halves for each report (self-report: 0.79; parents report: 0.82–0.85; cf. Table 3).

#### 3.2.5. Concordance

With respect to concordance, the intraclass-correlation coefficients between self-report and parents report scores for overall scale and sub-scales varied between 0.53 and 0.67 (cf. Table 4).

#### 3.2.6. Validity

Validity of the CHC-/YHC-SUN-SF measures can be assumed, e.g., as indicated by the associations with a single item on general health care satisfaction (self-report: 56; parents report: 0.75). As expected (cf. Table 5), moderate to small associations could be observed for self-reported health care satisfaction with self-reported general quality of life (EUROHIS-QOL: 0.35) and health-related quality of life (SF-8: 0.25/0.29).

## 4. Discussion

We aimed to investigate the psychometric properties of a newly developed short-form of the CHC-/YHC-SUN measure (CHC-/YHC-SUN-SF), assessing satisfaction with health care in children and adolescents with chronic conditions. We analyzed reliability, factorial validity, as well as convergent and divergent validity for both versions (self-report, parents report) of the measure.

With respect to the factorial structure of the CHC-/YHC-SUN-SF with one global factor (satisfaction with health care) and three sub-ordinated factors, our analysis confirmed the model for both versions independently (self-report, parents report). All fit indices reached values in a very good range, except for the RMSEA of the parents report version. Moreover, as compared to alternative measurement models, the model with this factor structure fitted best to the data. Given the loadings on the higher order factor, analyses supported the assumption that the computation of both a global score as well as three subscale scores of health care satisfaction is meaningful. Despite their shortness, these subscales are psychometrically justified in principle. However, the total score should be used as the standard indicator to document CHC-/YHC-SUN-SF study results. 

Reliabilities in terms of internal consistencies for the overall scale with six items (and its test-halves with three items each) as well as for the three sub-scales with two items each reached high values for both versions. Scores for both versions showed a moderate association, which is in line with the literature on self-/parents agreement for patient-reported outcome measures in health care research. Empirical findings reported by Riley (2004) point to the fact that in pediatric care, adolescents as well as their parents are able to give accurate assessments according to their individual perspectives.

Convergent validity of the CHC-/YHC-SUN-SF measures can be assumed, as indicated by associations with a single-item measure on general health care satisfaction, indicating both, shared variance reflecting sufficient overlap in assessing the same underlying construct (satisfaction) as well as sufficient specificity accounting for different approaches to satisfaction assessment (domain-related vs. generic).

Divergent validation of the CHC-/YHC-SUN-SF measures revealed that the relationships between satisfaction with health care and quality of life were widely confirmed for both instruments. Whereas correlation coefficients between satisfaction with care and general quality of life (EUROHIS-8) ranged between 0.26 and 0.35, correlation coefficients with health-related quality of life (SF-8) proved to be even lower (r = 0.17–0.29).

Despite these sound psychometric properties of the measure, several limitations of the validation study must be taken into consideration: Our study covered only a limited age range, given that participants in this sample were adolescents and young people 14 years old or older only, but the questionnaire targeted both adolescents and children. Another issue is a potential bias, because response behavior regarding satisfaction with health care among patients who are willing to participate may be different from those that do not prefer to do so. Moreover, as for any short-form development, content validity decreased by significantly reducing the number of items. Additionally, we were unable to examine the instrument’s ability to detect change over time and test-retest reliability and did not test for known-groups validity as well as for invariance by sex, age, or other attributes. Finally, the analysis was restricted to data that were collected in German language only and therefore may reflect some characteristics specific to a German context, which limits the impact of our study. Previous research has shown differences in pediatric health care provision between different European countries, e.g., for adolescents with disorders/differences of sex development [28]. Due to these limitations, further studies with improved study designs and methods are needed to provide evidence that the newly developed short-forms could be applied to other populations and age groups as well.

Future research options in the field of pediatric health services research fostered by this short instrument cover a wide range of topics. First, the measure provides opportunities to study the association between adolescents’ self-assessment and parents’ assessment regarding satisfaction with health care of the young patients. Moreover, considering the multidimensionality of the measure, it fosters comparisons between objective indicators of health care services and subjective satisfaction with health care, as almost demonstrated [28]. Finally, both short-form versions should be validated in other languages, especially for those versions almost available for the original CHC-/YHC-SUN measures. Therefore, we envisage the further validation of the CHC-/YHC-SUN-SF in an international study investigating cross-cultural comparability in terms of measurement invariance across different languages and countries. This would enable researchers to compare satisfaction with heath care between different countries. Variations across countries might then in part reflect different organization and financing of health care for youth with chronic conditions.

## 5. Conclusions

In summary, both versions of the CHC-/YHC-SUN-SF have many strengths that demonstrate the relevance of this successful validation. Beside its psychometric quality, the measure is economical and easy to administer. These are essentially important features for use in pediatric routine care. In addition, despite its brevity, it does not just provide a generic assessment of satisfaction with heath care only, but also delivers information about satisfaction with several core aspects of health services provision specifically and is applicable across a wide range of conditions. Taken this together, the CHC-/YHC-SUN-SF has huge potential in both pediatric health services research and pediatric clinical care.

## Figures and Tables

**Figure 1 ijerph-18-12196-f001:**
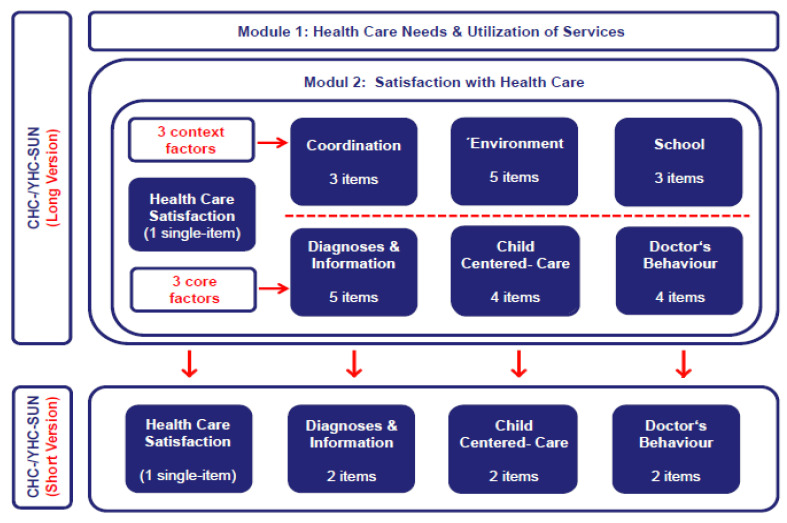
Overview of versions and content of different CHC-/YHC-SUN instruments for the assessment of special health care needs, health care utilization, and satisfaction with health care in children and adolescents with chronic health conditions.

**Table 1 ijerph-18-12196-t001:** (**a**) Descriptive item characteristics for the self-report of the CHC-/YHC-SUN-SF (*n* = 469). (**b**) Descriptive item characteristics for the parents report of the CHC-/YHC-SUN-SF (*n* = 370).

(a)					
Item	Domain	m	sd	Skewness	Curtosis
1.	Diagnosis/Information	3.47	1.01	−0.25	−0.44
2.	Diagnosis/Information	3.66	1.01	−0.36	−0.50
3.	Child-Centered Care	3.83	1.00	−0.57	−0.25
4.	Child-Centered Care	4.05	0.99	−0.91	0.32
5.	Doctor’s Behavior	4.00	0.96	−0.84	0.27
6.	Doctor’s Behavior	3.89	1.07	−0.77	−0.11
7.	(Generic Single-Item-Measure)	3.68	1.00	−0.34	−0.52
(**b**)					
**Item**	**Domain**	**m**	**sd**	**Skewness**	**Curtosis**
1.	Diagnosis/Information	3.41	1.06	−0.44	−0.41
2.	Diagnosis/Information	3.59	1.07	−0.62	−0.16
3.	Child-Centered Care	3.94	0.92	−0.68	0.03
4.	Child-Centered Care	4.06	0.92	−0.89	0.43
5.	Doctor’s Behavior	3.90	0.89	−0.57	0.14
6.	Doctor’s Behavior	3.93	0.93	−0.73	0.22
7.	(Generic Single-Item-Measure)	3.80	0.88	−0.37	−0.30

**Table 2 ijerph-18-12196-t002:** (**a**) Selected Indices for Fit Statistics and standardized regression weights for items of Confirmatory Factor Analysis of the self-report version of the CHC-/YHC-SUN-SF (*n* = 469). (**b**) Selected Indices for Fit Statistics and standardized regression weights for items of Confirmatory Factor Analysis of the parents report version of the CHC-/YHC-SUN-SF (*n* = 370).

(a)			
Fit Statistics	Model 1	Model 2	Model 3
Chi-Square	111.34	67.570	14.48
df	9	7	6
p (Chi-Square) *	0.005	0.005	0.129
CFI	0.91	0.94	0.99
RMSEA	0.18	0.16	0.06
RMSEA (CI-90%)	(0.15–0.21)	(0.13–0.19)	(0.02–0.11)
P	0.000	0.000	0.246
SRMR	0.07	0.12	0.02
β _item 1_	0.67 (factor 1)	0.92 (factor 1)	0.86 (factor 1)
β _item 2_	0.65 (factor 1)	0.85 (factor 1)	0.81 (factor 1)
β _item 3_	0.81 (factor 1)	0.87 (factor 2)	0.82 (factor 2)
β _item 4_	0.85 (factor 1)	0.91 (factor 2)	0.88 (factor 2)
β _item 5_	0.80 (factor 1)	0.88 (factor 3)	0.80 (factor 3)
β _item 6_	0.80 (factor 1)	0.86 (factor 3)	0.82 (factor 3)
(**b**)			
**Fit Statistics**	**Model 1**	**Model 2**	**Model 3**
Chi-Square	286.61	55.77	23.42
df	9	7	6
p (Chi-Square) *	0.000	0.000	0.001
CFI	0.84	0.97	0.99
RMSEA	0.30	0.14	0.09
RMSEA (CI-90%)	(0.27–0.33)	(0.11–0.18)	(0.06–0.13)
P	0.000	0.000	0.000
SRMR	0.09	0.09	0.02
β _item 1_	0.68 (factor 1)	0.92 (factor 1)	0.91 (factor 1)
β _item 2_	0.66 (factor 1)	0.95 (factor 1)	0.88 (factor 1)
β _item 3_	0.91 (factor 1)	0.95 (factor 2)	0.93 (factor 2)
β _item 4_	0.90 (factor 1)	0.91 (factor 2)	0.93 (factor 2)
β _item 5_	0.87 (factor 1)	0.95 (factor 3)	0.91 (factor 3)
β _item 6_	0.84 (factor 1)	0.90 (factor 3)	0.86 (factor 3)

Notes. Sample sizes vary depending on missing data. p *: corrected *p*-values applying Bollen–Stine bootstrap method. CFI: Comparative Fit Index; RMSEA: Root-Mean-Square Error of Approximation; SRMR: Standardized Root-Mean Residual. β: Standardized regression weight.

**Table 3 ijerph-18-12196-t003:** Internal consistencies for the overall scale and subscales of the self-report and parents report versions of the CHC-/YHC-SUN-SF (*n* = 469).

CHC-/YHC-SUN-SF Score	Number of Items	Self-Report	Parents-Report
		Cronbach’s Alpha	Cronbach’s Alpha
Total	6	0.89	0.92
Diagnosis/Information	2	0.82	0.90
Child-Centered Care	2	0.84	0.93
Doctor’s Behavior	2	0.79	0.88

**Table 4 ijerph-18-12196-t004:** Intraclass-correlation coefficients between self-report and parents report scores for overall scale and sub-scales of the CHC-/YHC-SUN-SF (*n* = 355).

**CHC-/YHC-SUN-SF Score**	**ICC**	**CI-95%**
Total	0.67	0.60–0.73
Diagnosis/Information	0.53	0.43–0.62
Child-Centered Care	0.64	0.56–0.71
Doctor’s Behavior	0.56	0.46–0.64

**Table 5 ijerph-18-12196-t005:** Correlation coefficients between overall scale and sub-scale scores of the self-report version of the CHC-/YHC-SUN-SF (*n* = 469) and generic satisfaction with health care (single CHC-/YHC-SUN single-item measure) as well as several quality of life indicators (SF-8, EUROHIS-QOL; *n* = 469).

		Total	Diagnosis/Information	Child-Centered Care	Doctor’s Behavior
Divergent validity	M (SD)	r	r	r	r
SF-8 Somatic Sum Score	50.42 (6.37)	0.29 **	0.28 **	0.25 **	0.22 **
SF-8 Mental Sum Score	49.52 (6.31)	0.25 **	0.29 **	0.20 **	0.17 **
EUROHIS-QOL Total Score	4.07 (0.52)	0.35 **	0.35 **	0.31 **	0.26 **

Notes. ** *p* < 0.01; r = Pearson-correlation coefficient.

## Data Availability

Due to data protection issues, the data cannot be shared.

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
