# Peer review of "A Short Form of the Child/Youth Health Care Questionnaire on Satisfaction, Utilization, and Needs in Children and Adolescents with a Chronic Condition (CHC-SUN-SF/YHC-SUN-SF)"

_ijerph, 2021, doi:10.3390/ijerph182212196_

Round 1
Reviewer 1 Report
In my opinion the objective of the study is clear and well formulated. Likewise, the data analysis is precise and methodologically correct. Furthermore, both the discussion and the conclusions of the study are consistent with the results obtained. Nevertheless, although I consider that the study can be published in the terms in which it is written, I believe that it could be improved by including in the introduction the existing lines of research on the study of the satisfaction of child and adolescent patients, as well as the literature that highlights the importance of patient satisfaction in the recovery process, follow-up of prescriptions or adherence to treatment, and by addressing in more detail the differences between the evaluations of children and adolescents and those of their parents, and between the latter according to gender. But I reiterate that I believe this is optional. Finally, I congratulate you on your work and I hope that your instrument will find a place among the people working in this area of research. , I believe that the authors should introduce an appendix with the final version of the questionnaire and make a call to those researchers from other countries who would like to translate it into their respective languages and work on its validation in other cultural contexts. These improvements would add interest to the study and would increase the number of consultations.
Author Response
Dear Reviewer,
we would like to thank you for the swift review of our manuscript and your helpful comments. We have revised our submission to address your concerns, and we think that this has substantially improved our manuscript. Please find our responses to each of your comments attached. We have uploaded two versions of the revised manuscript, with and without tracked changes. We hope that you are satisfied with the changes we made.
Sincerely, Holger Muehlan, on behalf of all authors.

Reviewer 2 Report
General comments
This study aimed to develop and validate the short version of the Child/ Youth Health Care Questionnaire on Satisfaction, Utilization, and Needs. This instrument would be useful in understanding the health care satisfaction from patients’ point of view in the groups of children’s parents and youths. Although you have already written this manuscript, my major concerns over the study and/or the manuscript include Materials and Methods, Results, and Discussion.
Specific comments
Title: I would suggest a shorter title and delete the abbreviation to present what the research team did in this study.
Introduction: This part provided a clear background of the use of CHC-/YHC-SUN. One minor concern is the abbreviation, this issue also happened across the manuscript, such as PROM’s, PREM’s, HRQoL, DISQoL, etc. I would suggest using the full term if those terms do not exist frequently. Also, according to the original CHC-SUN, the abbreviation did not have the word “questionnaire”, you may consider deleting it across the manuscript.
Materials and Methods:
Line 97: What are SR and PR?
Line 100-101: Please use the full term of SF-8 and SF-36 when both are the first time to appear in the manuscript.
Line: 107: What are WHOQOL and WHOQOL-8?
Line 97-114: The authors should also provide the reliability and validity of all instruments and also when they are applied in different populations and groups.
Line 139: Please consider the cut-off value of CFI, IFI and TLI ≥ 0.9 (Bentler and Bonett, 1980), RMSEA ≤ 0.06 and SRMR ≤ 0.08 (Hu and Bentler, 1999).
Data samples: Although it was a series of cross-national projects, the data samples of this study were only from German and Austrian in German. The research team should clearly describe this situation or else, this may overestimate the impact of your study. In addition, has this study obtained ethical approval and parental consent?
Results:
Line 183-186, 191-193: Any reference(s) in this principle and psychometric properties?
Line 214-215: To my understanding, this study validates the CHC-SUN, however the age range of 14 – 24. Where are the results of the children group? Or else, this study can only be the validation of YHC-SUN-SF.
Discussion:
Line 280: RMSEA is a robust indicator of CFA. Since the parents' report is not fit in the model even using of higher-order factor method, how does your study justify the instrument?
Regarding the limitations, you should improve your study design and methods, and the presentation of results to justify your instrument can apply to more populations and age groups.
Author Response

(The authors gave the same response as above.)

Reviewer 3 Report
Thank you for asking me to review this article.
Investigating the determinants of child well-being regarding health care considered in their physical, psychological and social dimensions is fundamental in order to understand the general satisfaction with health care and the possible strategies to be implemented for a correct management of the chronic infant patient, guaranteeing the continuity of care, the quality of care and the integration of social and health interventions.
The work presents very interesting results, only minor revision is recommended in relation to the form: in the introductory paragraph, the objectives that guided the research question are clear, but the hypotheses that the research intended to verify could be deepened in order to make more clear the context. Furthermore, in lines 77-94 the authors refer to the details of the questionnaire and in lines 97-98 they refer to the background for the specifics of the same; in this regard, to make the reading smoother, the authors may think of describing the method chosen for the investigation in the "Materials and Method" section, while the introductory section could focus only on deepening the context and the objectives of the research.
Author Response

(The authors gave the same response as above.)
